# From Waste to Resources: Sewage Sludges from the Citrus Processing Industry to Improve Soil Fertility and Performance of Lettuce (*Lactuca sativa* L.)

Caterina Lucia, Daniela Pampinella *, Eristanna Palazzolo, Luigi Badalucco [ID] and Vito Armando Laudicina [ID]

Department of Agriculture, Food and Forest Sciences, University of Palermo, Viale delle Scienze, Building 4, 90128 Palermo, Italy; caterina.lucia@unipa.it (C.L.); eristanna.palazzolo@unipa.it (E.P.); luigi.badalucco@unipa.it (L.B.)
* Correspondence: daniela.pampinella@unipa.it; Tel.: +39-09123897074

**Abstract:** The citrus industry produces a large number of sludges as a consequence of citrus wastewater treatment. The correct disposal of citrus sewage sludges (CSSs) has been attempted using anaerobic digestion, aerobic digestion, and lime stabilization. However, since CSSs hold nitrogen, phosphorus, and other macronutrients required by crops, in line with the circular economy principles, they could be utilized for agricultural purposes, such as organic fertilizer. The aim of this study was to assess the effect of CSSs supplied at different doses on soil fertility and lettuce performance. To this end, a pot experiment was established. The soil was amended with CSSs at three different concentrations (2.5, 5, 10 t ha$^{-1}$). After 46 days of lettuce growth, the experiment was stopped, and soils and plants were analyzed. Soil amended with CSSs showed an increase in total organic C ranging from 7% to 11%. Additionally, available P increased but only at the highest CSS dose. The addition of CSSs affected the biochemical properties of soil, but a univocal trend related to the number of CSSs applied was not found. Microbial biomass C increased only with the highest dose of CSS applied, while the metabolic quotient ($qCO_2$) decreased. Such a positive effect on soil fertility and soil microorganisms, in turn, lead to an increase in lettuce biomass. Moreover, results indicated that following CSS addition, lettuce crops adsorbed more N in leaves than in roots, whereas P, Ca, Mg, K, and Na showed an opposite pattern and increased more consistently in roots. In conclusion, amendment with CSSs enhances soil fertility by increasing, regardless of CSS dose, total organic C, and, at the highest dose, P availability and microbial biomass C. Such improvement in soil fertility, in turn, increases lettuce biomass production without affecting its quality, i.e., alteration of the (K + Na)/(Ca + Mg) ratio.

**Keywords:** citrus sewage sludges reuse; organic amendment; circular economy; nutrient recovery

## 1. Introduction

Waste management is one of the biggest challenges facing contemporary society [1]. Waste management in the citrus processing industries is evolving into a significant concern due to several regulations [2,3]. The citrus industry produces more than 700 million m$^3$ of citrus wastewater (CWW) per year [4], thus posing significant environmental and financial challenges [5]. Indeed, the disposal unit cost of CWW is about 44 € m$^{-3}$ (FAO, 2016). This suggests that citrus processing industries should move to a strategic approach concerning proper CWW treatment and disposal, according to laws and regulations, in order to minimize the environmental and economic charges [5,6].

According to EU legislation [7], CWW can be released into urban collecting systems after appropriate treatments. Among the different approaches to treat CWW, intensive biological treatment is the most developed [8]. A consequence of the biological treatment of CWW is the excess generation of sludge. Corsino et al. [9] estimated that the aerobic approach used to treat CWW produces between 0.10 and 0.30 kg of sludge per kg of



chemical oxygen demand (COD). Given that the average COD content in CWW is between 5 and 27 kg COD m$^{-3}$, the specific productivity of sludge may range between 0.5 and 9.0 kg sewage sludge m$^{-3}$ of treated wastewater. The excess of sludge generated from the transformation of the organic matter into new biomass entails high expenditure for its treatment and disposal [10], which may account for 30–40% of the total capital cost and 50% of plant operating costs [10,11]. Thus, innovative approaches focused on decreasing the surplus sludge generation or valorizing the sludge once created should be taken into consideration to minimize the impact of sludge treatment on the overall operating costs [12]. Anaerobic digestion processes have been widely employed to reduce the volume and weight of sludge and to produce biogas, despite their deficiencies [13]. It has been estimated that anaerobic digestion is used in 50% of sewage treatment facilities in the EU, whereas aerobic digestion is used in 18%, and lime stabilization accounts for 4%. Sludge stabilization is not used in 24% of facilities [14]. Other resourceful applications of sludges can be their conversion into biochar and the following application to the soil to improve chemical and biochemical soil fertility. Indeed, some recent studies [15–17] have found that biochar, obtained from sewage sludge, in combination with phosphorus solubilizing bacteria improved the stable form of toxic metal minerals and microbial abundance in lead-/cadmium-contaminated soil. An alternative management/treatment approach may be the reuse of treated biological sewage sludge as an organic amendment to improve soil fertility [18]. Sludge could be utilized as fertilizer since crops utilize nitrogen and phosphorus following the mineralization process of the held organic matter [19]. Indeed, the most popular technique for treating and disposing of sewage sludge in China is anaerobic digestion/aerobic composting, as well as land application [20]. In accordance with the circular bio-economy concept, agricultural use may be a workable solution given the high organic content of the sludge resulting from CWW treatment [21]. There is a large volume of published studies describing the role of sewage sludge in agriculture. According to Jamil et al. [22], the addition of different levels of domestic sewage sludge to soil increased its content in organic matter by about 29% and macronutrients, such as N and P, by about 433%, while K increased by 23%. In a study conducted by Hamdi et al. [23], it was shown that the addition of urban sewage sludge (120 t ha$^{-1}$) increased the content of N, P, and K, respectively, by 0.18%, 246, and 22 mg kg$^{-1}$ in sandy loam soil, and by 0.15%, 356, and 19.2 mg kg$^{-1}$ in sandy soil. A different trend has been reported regarding soil microbial biomass following the application of sewage sludge (SS). According to Franco-Otero et al. [24], applications of SS did not affect significantly soil microbial biomass compared to soil that had not been amended. Additionally, Picariello et al. [25] did not find any positive effect on soil microbial biomass following the amendment with SS. However, this behavior depends on the amount of sludge applied to the soil.

However, sewage sludge may contain a variety of hazardous and dangerous compounds, including heavy metals [26]. The pH, cation exchange capacity, organic matter content, and mobility and form of certain metals all have an impact on the release of heavy metals present in sewage sludge, as well as their adsorption and speciation in soil [27].

Sludge should be stored since there are technical issues coming from the fact that it is produced all year long but applied to land only occasionally. Nevertheless, there are also societal and technological barriers [28,29] to the utilization of sewage sludge for agricultural applications. Sludge from the agro-food industry may be applied to soils in doses not exceeding 15 t ha$^{-1}$ of dry matter over a three-year period [30].

Although different approaches have been carried out for proper citrus sewage sludge (CSS) disposal, to our knowledge, no studies are reported in the literature concerning the effect of their application to soil on fertility and crop performance. Thus, the aim of this study was to assess the effect of CSSs on soil fertility and lettuce (*Lactuca sativa* L.) performance.

## 2. Materials and Methods

### 2.1. Experimental Set-Up

The experiment was carried out in a greenhouse located at the Department of Agricultural, Food, and Forest Sciences of the University of Palermo (Italy). About 50 kg of soil, collected at a 0–20 cm depth from an agricultural area of the Department of Agricultural, Food, and Forestry Sciences, University of Palermo, were air dried, sieved at <4 mm, and mixed with 10% of peat (*v/v*). The main characteristics of the soil after peat amendment were clay 27%, sand 51%, silt 22%, pH and electrical conductivity in water extract (1:2.5, *w/v*) 7.9 and 0.48 dS m$^{-1}$, respectively, total organic carbon 48.5 g kg$^{-1}$, total nitrogen 3.5 g kg$^{-1}$, and total carbonates 17%. Soil aliquots of 500 g were mixed with 1, 2, and 4 g of CSSs corresponding to an amount of 2.5, 5, and 10 t ha$^{-1}$ (CSS2.5-P, CSS5-P, and CSS10-P, respectively). After that, plastic pots (10 cm × 10 cm × 12 cm) were filled with 500 g of soil, and *Lactuca sativa* L. was planted (one plant at two leaf stages per pot). The growth experiment lasted 46 days. During this timeframe, plants were watered daily with tap water. Three replicates per treatment were run. A control treatment was also prepared (CTR, no addition of CSSs) (Figure 1).

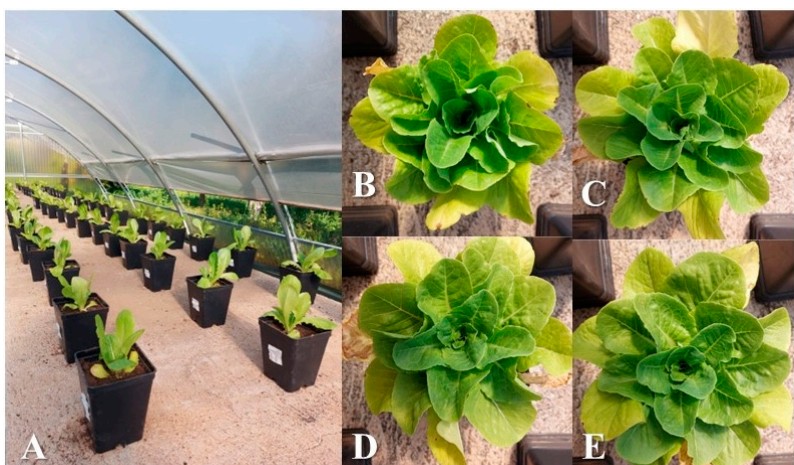

**Figure 1.** (**A**) Growth experiment with lettuce plants cultivated in soil amended with (**B**) no sludge (CTR), (**C**) 2.5 t ha$^{-1}$ (CSS2.5-P), (**D**) 5 t ha$^{-1}$ (CSS5-P), and (**E**) 10 t ha$^{-1}$ (CSS10-P).

### 2.2. Citrus Sewage Sludges

CSSs were collected from a citrus processing industry (Agrumaria Corleone spa, Palermo, Italy). These consisted of the waste produced by the wastewater treatment process of the citrus industry [5]. CSSs were characterized to determine their main chemical properties. Reaction and electrical conductivity (EC) (1:2.5, *w/v*) were determined by a pHmeter (FiveEasy, Mettler Toledo Spa, Milan, Italy) and a conductometer (HI5321, Hanna Instruments Italia srl, Padua, Italy), respectively. Total nitrogen (TN) was determined by the Kjeldahl method [31], while total organic carbon (TOC) was determined by the Walkley–Black wet oxidation method [32]. Total phosphorus (Total P) was determined by the Spectroquant® Phosphate test using a spectrophotometer (UVmini-1240, Shimadzu Italia srl, Milan, Italy) after the formation of an orange-yellow complex. The content in macronutrients and heavy metals was determined by MP-AES (Agilent 4210 MP-AES, Milan, Italy) following CSS mineralization by wet digestion procedure (HNO$_3$ and 30% H$_2$O$_2$). Briefly, 0.5 g of CSSs were weighed into a porcelain crucible and placed in a cool muffles furnace. The furnace temperature was set to reach 500 °C in about 2 h and samples were left to stand overnight. Then, samples were removed and after cooling, 5 mL of HNO$_3$ and 5 mL of 30% H$_2$O$_2$ were added. Then, they were placed in a hot plate at 250 °C for 2 h and, if needed, the addition of HNO$_3$ and 30% H$_2$O$_2$ was repeated After cooling, the digested sample was transferred quantitatively to a volumetric flask and brought to a

20 mL volume with 2% $HNO_3$ [33]. As and Hg were not determined since they were absent in the raw materials.

### 2.3. Soil Analyses

At the end of the 46 d growth period, plants were removed for further analysis. The soil of each pot was fully mixed, air dried, sieved at <2 mm, and then analyzed. Soil reaction, electrical conductivity, and total nitrogen were determined similarly to CSSs. Total carbonates were determined gas volumetrically using the Dietrich-Frühling calcimeter [34]. The exchangeable basic cations in soil ($Ca^{2+}$, $Mg^{2+}$, $Na^+$, and $K^+$) were determined on soil extract with 1 M ammonium acetate by MP-AES (Agilent 4210 MP-AES).

An aliquot of dried soil samples, moistened up to 50% of their water-holding capacity (WHC), was used for the determination of microbial biomass C and respiration ($CO_2$ emission). Microbial biomass C (MBC) was determined by the fumigation-extraction method [35,36]; it corresponded to the difference between organic C extracted by 0.5 M $K_2SO_4$ from $CHCl_3$-fumigated and not fumigated samples, multiplied by 2.64.

Soil $CO_2$ emission was assessed by incubating 20 g of soil at 50% WHC in 200 mL glass jars sealed with rubber stoppers holding silicone septa and placed in the dark at 22 °C for 10 days [37]. The $CO_2$ accumulated after 3 days of incubation in the headspace of the glass jars was determined by injecting 1 mL of air from each jar into a gas chromatograph (TraceGC, Thermo Fisher Scientific Inc., Waltham, MA, USA) equipped with a thermal conductivity detector and 80–100 mesh stainless-steel columns packed with Poropak Q using He as the carrier.

### 2.4. Plant Analyses

One day before removing lettuce plants, the SPAD index was measured by a chlorophyll meter (Konica Minolta SPAD-502), and results were expressed as the average of two readings on two leaves for each plant. After removing the lettuce plants, the roots were first washed with tap water and then with distilled water. Roots and leaves were separated, oven-dried at 60 °C until a constant weight (ca. after 48 h), and separately weighed. Dried roots and leaves were ground and kept in a plastic bottle at 4 °C before further analysis. Total nitrogen was determined by the Kjeldahl digestion method [31]. Total P was determined, on mineralized plant samples, by acid ($HNO_3$ and 30% $H_2O_2$) and the wet digestion procedure [33] and by the Spectroquant® Phosphate test using a spectrophotometer (UVmini-1240, Shimadzu Italia srl, Milan Italy) after the formation of an orange-yellow complex. The total content of Ca, Mg, K, and Na was determined, by MP-AES (Agilent 4210 MP-AES) on mineralized plant samples. The wet digestion procedure applied was the same used for CSS samples. Reagents used for CSS, soil, and plant analyses were of ACS grade. Deionized water was used for the preparation of the extractant solution. QA/QC measures included an analysis of procedural blanks and replicates of standard solutions.

## 3. Results

### 3.1. Chemical Characteristics of CSSs

The chemical characteristics of CSSs used in this study are reported in Table 1. CSSs had a subalkaline pH and high electrical conductivity, although it was lower than 4 dS m$^{-1}$, the threshold for saline soils. CSSs had a high content of total N, so the C/N ratio was about 6. Among macronutrient cations, Ca was the most abundant, followed in order by Mg, P, K, and Na. Pb, Cr, and Cd were not detectable, whereas the concentration of the other metals was lower than the legal limits, according to Directive 86/278/ECC [38].

**Table 1.** Main chemical parameters determined on citrus sewage sludges (CSSs) according to Directive 86/278/ECC [38]. n.d., not detectable.

| Parameter | (CSSs) |
|---|---|
| pH | $7.9 \pm 0.2$ |
| E.C. (dS m$^{-1}$) | $3.72 \pm 0.06$ |
| TOC (g kg$^{-1}$) | $255.1 \pm 1.3$ |
| TN (g kg$^{-1}$) | $40.0 \pm 1.7$ |
| P (g kg$^{-1}$) | $2.7 \pm 0.2$ |
| Ca (g kg$^{-1}$) | $14.7 \pm 0.6$ |
| K (g kg$^{-1}$) | $2.9 \pm 0.7$ |
| Na (g kg$^{-1}$) | $2.3 \pm 0.3$ |
| Mg (g kg$^{-1}$) | $4.6 \pm 0.2$ |
| Zn (g kg$^{-1}$) | $0.20 \pm 0.01$ |
| Cd (g kg$^{-1}$) | n.d. |
| Fe (g kg$^{-1}$) | $1.3 \pm 0.1$ |
| Cu (g kg$^{-1}$) | $0.02 \pm 0.01$ |
| Ni (g kg$^{-1}$) | $0.01 \pm 0.00$ |
| Pb (g kg$^{-1}$) | n.d. |
| Mn (g kg$^{-1}$) | $0.04 \pm 0.00$ |
| Al (g kg$^{-1}$) | $0.30 \pm 0.04$ |
| Cr (g kg$^{-1}$) | n.d. |

*3.2. The Effect of CSSs on Chemical and Biochemical Soil Properties*

The addition of CSSs increased, although not significantly, pH by 0.3 in soil amended at the highest dose, thus reaching the value of 7.5 (Table 2). On the contrary, electrical conductivity decreased from 1.2 to 0.9 dS m$^{-1}$ in all amended soils, regardless of the number of CSSs supplied. Total organic C increased following the addition of CSSs by about 9%, on average. On the other hand, total N decreased compared to CTR when CSSs were added at the lowest dose, while it did not change with the two highest ones. Therefore, the TOC/TN ratio increased following the addition of CSSs at the two lowest doses. Available P decreased with the lowest CSS dose and increased only with the highest one.

**Table 2.** Chemical properties determined on soil samples after the addition of a different amount (2.5 t ha$^{-1}$, 5 t ha$^{-1}$, 10 t ha$^{-1}$) of citrus sewage sludge (CSS).

| Treatment | pH (1) | EC (2) (dS m$^{-1}$) | TOC (3) (g kg$^{-1}$) | TN (4) (g kg$^{-1}$) | TOC/TN | Available P (mg kg$^{-1}$) |
|---|---|---|---|---|---|---|
| CTR | $7.2 \pm 0.2$ ab | $1.18 \pm 0.08$ a | $57.7 \pm 1.2$ b | $3.4 \pm 0.2$ a | $16.9 \pm 0.7$ b | $98.8 \pm 0.6$ b |
| CSS2.5-P | $7.1 \pm 0.0$ b | $0.89 \pm 0.04$ b | $62.5 \pm 3.0$ a | $3.1 \pm 0.1$ b | $19.9 \pm 1.3$ a | $83.2 \pm 0.6$ c |
| CSS5-P | $7.2 \pm 0.0$ ab | $0.90 \pm 0.03$ b | $64.0 \pm 1.3$ a | $3.3 \pm 0.1$ ab | $19.4 \pm 0.8$ a | $102.0 \pm 2.3$ b |
| CSS10-P | $7.5 \pm 0.4$ a | $0.91 \pm 0.06$ b | $61.8 \pm 1.0$ a | $3.4 \pm 0.1$ ab | $18.3 \pm 0.4$ ab | $115.0 \pm 7.6$ a |

(1) pH in water 1:2.5, *w/v*; (2) EC, electrical conductivity 1:5, *w/v*; (3) TOC, total organic carbon; (4) TN, total Kjeldahl nitrogen. Reported values are means $\pm$ standard deviations (n = 3). Different letters indicate significant differences among treatments at *p* < 0.05.

Na and K were not affected by the addition of CSSs. On the other hand, Ca, Mg, Cu, and Al decreased compared to the not amended soil, whereas among micronutrients, only Fe increased, regardless of the amount of CSS supplied (Table 3).

**Table 3.** Exchangeable basic cations in soil ($Ca^{2+}$, $Mg^{2+}$, $Na^{+}$, and $K^{+}$) determined on soil amended with different concentrations of CSS (2.5 t ha$^{-1}$, 5 t ha$^{-1}$, 10 t ha$^{-1}$).

| Treatment | $Ca^{2+}$ g kg$^{-1}$ | $Mg^{2+}$ g kg$^{-1}$ | $Na^{+}$ g kg$^{-1}$ | $K^{+}$ g kg$^{-1}$ |
|---|---|---|---|---|
| CTR | 11.36 ± 0.27 a | 0.870 ± 0.003 a | 0.306 ± 0.006 a | 0.309 ± 0.010 a |
| CSS2.5-P | 10.54 ± 0.02 d | 0.835 ± 0.007 b | 0.258 ± 0.006 a | 0.303 ± 0.011 a |
| CSS5-P | 10.92 ± 0.07 b | 0.827 ± 0.008 b | 0.273 ± 0.004 a | 0.297 ± 0.004 a |
| CSS10-P | 10.68 ± 0.05 c | 0.796 ± 0.006 b | 0.273 ± 0.010 a | 0.297 ± 0.018 a |

Reported values are means ± standard deviations (n = 3). Different letters indicate significant differences among treatments at $p < 0.05$.

The only biochemical soil property clearly affected by the addition of CSSs was the microbial biomass C, which, at the highest CSS dose, increased by 17%. The soil respiration ($CO_2$ emission) slightly decreased, regardless of the CSS dose applied, similar to the microbial quotient. The metabolic quotient was not affected by CSS (Table 4).

**Table 4.** Biochemical properties of soils amended with different concentrations of CSS (2.5 t ha$^{-1}$, 5 t ha$^{-1}$, 10 t ha$^{-1}$).

| Treatment | MBC (1) mg kg$^{-1}$ | $CO_2$ mg $CO_2$-C kg$^{-1}$ d.s. | Qmicr (2) % | qCO$_2$ (3) mg $CO_2$-C g$^{-1}$ MBC h$^{-1}$ |
|---|---|---|---|---|
| CTR | 786 ± 15 b | 31 ± 0.0 a | 1.5 ± 0.0 a | 1.5 ± 0.2 ab |
| CSS2.5-P | 743 ± 5 b | 29 ± 0.9 b | 1.2 ± 0.1 b | 1.3 ± 0.1 ab |
| CSS5-P | 771 ± 24 b | 30 ± 0.5 ab | 1.2 ± 0.1 b | 1.6 ± 0.0 a |
| CSS10-P | 920 ± 21 a | 29 ± 0.5 b | 1.5 ± 0.0 a | 1.3 ± 0.0 b |

(1) MBC, microbial biomass carbon; (2) Qmicr, microbial quotient; (3) qCO$_2$, metabolic quotient. Reported values are means ± standard deviations (n = 3). Different letters indicate significant differences among treatments at $p < 0.05$.

### 3.3. Lettuce Response to CSS Addition

The addition of CSSs at any dose to soil increased the plant's dry weight compared to the control by about 33% at the highest dose (Figure 2). Additionally, the chlorophyll index (SPAD) proportionally increased by increasing the CSS dose, and at the highest CSS dose, it was 22% higher than the control (Figure 3).

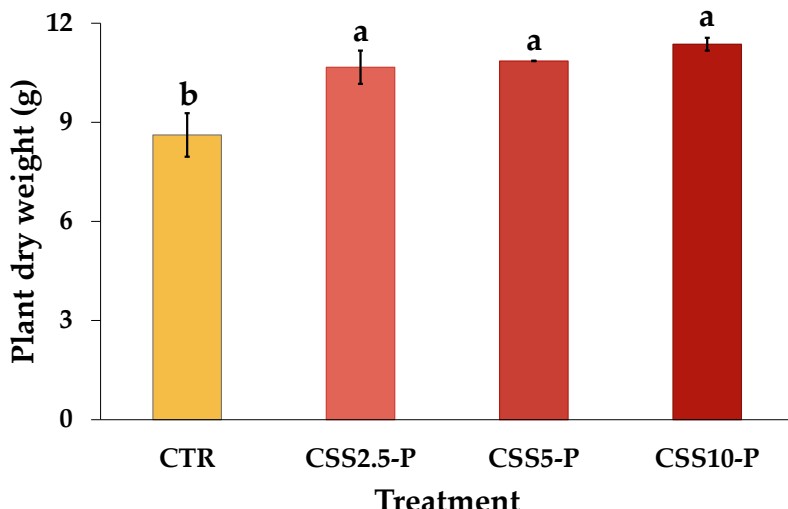

**Figure 2.** Plant dry weight of lettuce in soil amended with different doses of CSS: control (CTR), 2.5 t ha$^{-1}$ (CSS2.5-P), 5 t ha$^{-1}$ (CSS5-P), and 10 t ha$^{-1}$ (CSS10-P). Different letters indicate significant differences among treatments at $p < 0.05$.

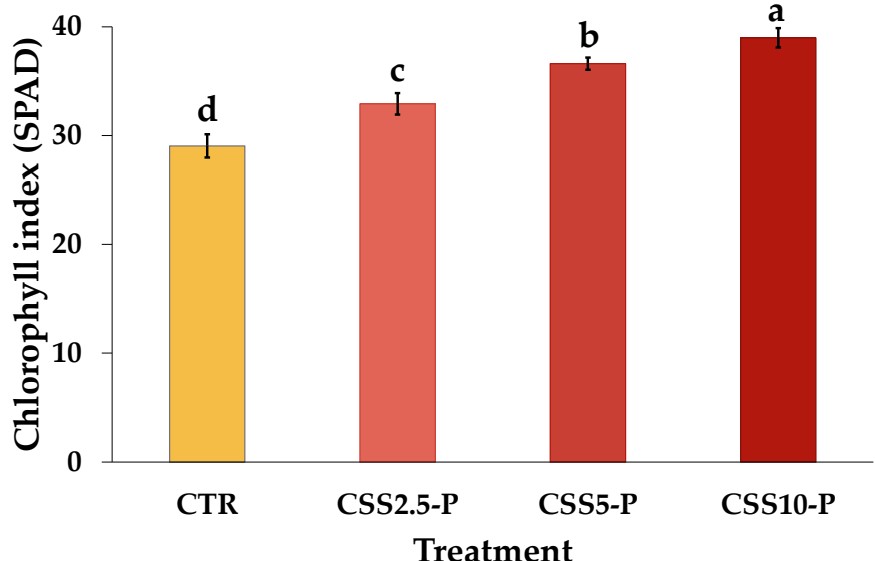

**Figure 3.** Chlorophyll index (SPAD) of lettuce in soil amended with different doses of CSS: control (CTR), 2.5 t ha$^{-1}$ CSS2.5-P, 5 t ha$^{-1}$ (CSS5-P), and 10 t ha$^{-1}$ (CSS10-P). Different letters indicate significant differences among treatments at $p < 0.05$.

Generally, the total content of each investigated element in total plant tissue proportionally increased by increasing the number of CSSs supplied, although a different assimilation behavior was found between leaves and roots (Table 5).

**Table 5.** Macronutrient concentration in leaves and roots after treatment with CSSs at different doses and total content in total plant tissue.

| Leaves | CTR | CSS2.5-P | CSS5-P | CSS10-P |
|---|---|---|---|---|
| TN (g kg$^{-1}$) | 7.9 ± 0.1 c | 8.5 ± 0.3 b | 10.1 ± 0.9 a | 13.0 ± 3.7 a |
| P (g kg$^{-1}$) | 1.5 ± 0.2 a | 0.4 ± 0.0 b | 1.6 ± 0.2 a | 1.8 ± 0.2 a |
| Ca (g kg$^{-1}$) | 11.5 ± 0.2 a | 11.9 ± 0.4 a | 9.6 ± 0.8 b | 11.0 ± 0.8 ab |
| K (g kg$^{-1}$) | 10.0 ± 0.3 b | 10.9 ± 0.6 ab | 11.7 ± 0.8 a | 11.6 ± 0.8 a |
| Mg (g kg$^{-1}$) | 2.4 ± 0.3 a | 2.5 ± 0.4 a | 2.7 ± 0.4 a | 3.0 ± 0.5 a |
| Na (g kg$^{-1}$) | 2.9 ± 0.2 b | 2.8 ± 0.1 b | 3.8 ± 0.3 a | 3.8 ± 0.2 a |
| **Roots** | | | | |
| TN (g kg$^{-1}$) | 8.4 ± 0.5 ab | 7.7 ± 0.1 b | 9.3 ± 0.4 a | 9.0 ± 0.2 a |
| P (g kg$^{-1}$) | 1.7 ± 0.0 b | 1.1 ± 0.1 c | 2.3 ± 0.1 a | 2.3 ± 0.2 a |
| Ca (g kg$^{-1}$) | 6.9 ± 0.3 d | 15.5 ± 1.5 c | 39.1 ± 3.8 b | 66.4 ± 9.1 a |
| K (g kg$^{-1}$) | 1.9 ± 0.1 c | 3.8 ± 0.1 b | 4.9 ± 0.7 a | 4.2 ± 0.2 a |
| Mg (g kg$^{-1}$) | 3.2 ± 0.2 d | 4.4 ± 0.4 c | 9.1 ± 0.5 b | 11.2 ± 0.0 a |
| Na (g kg$^{-1}$) | 3.2 ± 0.3 b | 2.2 ± 0.2 c | 5.0 ± 0.1 a | 5.2 ± 0.1 a |
| **Total** | | | | |
| TN (mg) | 69 ± 6 d | 87 ± 5 c | 106 ± 9 b | 128 ± 12 a |
| P (mg) | 13 ± 1 b | 7 ± 1 c | 21 ± 2 a | 23 ± 2 a |
| Ca (mg) | 90 ± 7 d | 142 ± 13 c | 246 ± 18 b | 392 ± 27 a |
| K (mg) | 71 ± 6 b | 87 ± 9 ab | 95 ± 8 a | 96 ± 11 a |
| Mg (mg) | 22 ± 3 d | 35 ± 4 c | 60 ± 5 b | 74 ± 8 a |
| Na (mg) | 26 ± 3 b | 27 ± 2 b | 47 ± 6 a | 50 ± 6 a |

Reported values are means ± standard deviations (n = 3). Different letters indicate significant differences among treatments at $p < 0.05$.

Total N and P determined on both leaf and root samples showed an opposite pattern. Total N proportionally increased by increasing the amount of CSS supplied in the leaves, whereas no effect was found on roots. On the contrary, total P generally increased by increasing the amount of CSS supplied in roots, while no effect was found on leaves. Except for K, regardless of the treatment, nutrients and heavy metals were more concentrated in roots than in leaves.

The concentration of Ca, Mg, and Na in plant roots increased by increasing the amount of CSS supplied at the two highest doses. The concentration of the same metals in leaves did not differ compared to the control, except for Na, which increased from 2.9 to 3.8 g kg$^{-1}$. Potassium was more concentrated in leaves compared to roots. However, by increasing the amount of CSS supplied, it increased both in roots and leaves, and such an increase was more consistent in the former than in the latter.

## 4. Discussion

### 4.1. CSS Effect on Soil Fertility

Wastewater treatment produces a large number of sludges with different characteristics depending on the techniques used [5]. In general, sewage sludge holds organic substances, macronutrients, a variety of micronutrients, non-essential trace metals, organic microcontaminants, and microorganisms [39]. The organic components of sewage sludge have considerable soil conditioning qualities, while the macronutrients act as a good supply for plant growth [40]. The application of sewage sludge to soil allows nutrient recycling and may avoid or reduce the use of fertilizer in agriculture [41], which is in line with EU policy to improve the circular economy. Indeed, CSSs hold plant nutrients such as N, P, Ca, Mg, K, and other micronutrients, in addition to organic matter.

One of the crucial factors concerning the land application of sludge is the possible presence of heavy metals and their concentration. Some of these metals work with different enzymes as co-factors. However, they may be hazardous to both plants and microbe metabolism if they accumulate in the soil. The concentration of heavy metals in the sludge depends on the potential of the treatment plant and the type of wastewater, such as domestic or industrial [42]. However, the heavy metal concentration in CSS was below the maximum allowed by Directive 86/278/ECC [38], thus encouraging their reuse for agriculture purposes.

The addition of CSSs did not affect soil pH. On the other hand, electrical conductivity decreased, probably due to nutrient assimilation by plants according to the results reported in Table 3.

Soils supplied with CSSs showed an increase in organic C from 7% to 11%. Considering that organic C supplied by CSSs ranged from 0.9 to 3.5%, it can be stated that organic C increased due to rhizodeposition. Such an increase is reasonable and agrees with previous findings by Arif et al. [43] and Dhanker et al. [44], who investigated the effect of sewage sludge addition to soil and reported similar results. Due to the key role played by soil organic C in improving soil fertility, especially in arid and semiarid environments [45], such results suggest that CSSs can contribute to counteracting the decline of soil organic C and hence, soil fertility.

Nitrogen and phosphorous found in sewage sludge contribute to increasing soil fertility, and such elements are of primary importance for plant growth [46]. However, following the addition of CSSs, the total N in soil did not increase. Such results may be due to many reasons, such as the low amount of N added, N assimilation by both plants and soil microbes upon mineralization and ammonification processes, or N leaching.

On the other hand, available P increase, at the highest dose, suggests that P supplied by CSSs was greater than requested by plants or underwent immobilization/precipitation processes. The latter consideration is reasonable due to the soil pH and the high amount of Ca added by CSSs. The lack of available P increase at the two lowest doses, indeed, may be due to P assimilation by plants and/or by microorganisms. In fact, MBC increased at the

highest dose of CSSs supplied. Such results agree with what was reported by Richardson and Simpson [47], who suggested that soil microorganisms mediated P availability.

Despite $Ca^{2+}$, $Mg^{2+}$, $Na^{+,}$ and $K^+$ being supplied by CSSs, their content did not change compared to the control. This finding may be ascribed to plant adsorption or leaching, especially for the monovalent cations, following the addition of irrigation water.

Microbial biomass C and the parameters related to it play a crucial role in assessing soil quality and biological stress [48,49]. For example, the stress of soil microorganisms or disturbance, and consequently its metabolic efficiency, may be revealed by the $qCO_2$ [49,50]. With regard to such parameters, following the addition of CSSs, only a few differences between the treatments and the control were assessed. Microbial biomass C increased only by the highest dose of CSS, whereas $qCO_2$ decreased. Such results suggest that CSSs stimulate soil microorganisms' growth and improve their ability in utilising organic C substrates. The slight decrease in $qCO_2$ at the highest CSS dose disagreed with Fernandes et al. [51] and Armenta et al. [52], who reported an increase in $qCO_2$ by increasing the number of urban sludges supplied. $qCO_2$ has received many criticisms about its usefulness to evaluate soil microorganism stress. Notwithstanding, it is constantly used. The different response of $qCO_2$ to sludge addition may be due to the type of sludge that, in turn, affects C availability to soil microorganisms and microbial community structure [45,53]. The latter finding, thus, needs further investigation.

### 4.2. CSS Effect on Lactuca sativa

The improvement of soil fertility following the addition of CSSs had a positive impact on the biometric and chemical parameters of lettuce. Indeed, total dry weight increased by 25% compared to the control. Such an increase is ascribed to the nutrients supplied by CSSs and, in particular, to N and P.

Nitrogen is a fundamental constituent of proteins and chlorophyll; it plays a major role in plant and microbial growth. The rate of photosynthesis in plants and the cycling of carbon are negatively impacted by the lack of nitrogen and phosphorus [54]. Indeed, the high amount of N assimilated by the plant, in turn, increased the chlorophyll index (SPAD). These results are in line with those of Gattullo et al. [55], who found a similar value of the SPAD index in lettuce leaves. Moreover, such results confirm what was previously stated, i.e., organic N supplied by CSS was assimilated by plants, upon ammonification and nitrification, and thus does not accumulate in soil. Total N in leaves varied significantly compared to its content in roots. Indeed, it increased considerably in leaves by 64% compared to the control. These results agree with those obtained by Sönmez and Bozkurt [56], who reported an increase in total N in lettuce leaves following sludge application. Additionally, Khaliq et al. [57] reported an increase in total N that was greater in leaves than in roots. P content found in plants was of the same order of magnitude as that reported in other studies, e.g., González-Ponce et al. [58]. The increase in total P in plants at the two highest doses suggests that P supplied by CSSs was assimilated. However, assimilated P was not translocated from roots to leaves, as P concentration increased only in roots and not in leaves.

Additionally, greater amounts of Ca, Mg, K, and Na were adsorbed by plants as the concentration of CSSs supplied increased. Such elements were more concentrated in roots than in leaves. Whereas the concentration of all metals increased in roots, only K and Na slightly increased in leaves compared to the control. Such differential assimilation of metals is likely due to their valence, as divalent and trivalent cations slowly translocated from roots to leaves. However, the addition of CSSs did not affect the ratio between (K + Na) and (Ca + Mg), and hence lettuce quality [59].

## 5. Conclusions

The citrus industry produces a large number of sludges as a consequence of citrus wastewater treatment. Due to a lack of specific guidelines and a methodology for applying the correct sludge management system, as well as the significant investment and mod-

ernization costs for obsolete sewage treatment plants, the management of this waste is still an issue. However, in line with the circular economy principles, the use of CSSs for agricultural purposes should be more than desirable.

The results of this study suggest that CSSs applied to soil as organic amendments improve soil fertility by supplying nitrogen, phosphorus, and other macronutrients. The positive effect on soil fertility, in turn, leads to an increase in lettuce yield without affecting its quality with regard to the (K + Na)/(Ca + Mg) ratio.

However, further studies are needed to establish the proper number of CSSs to be applied and their potential as possible fertilizers to support other crops. Moreover, to develop a full picture of CSS agricultural reuse, additional studies will be needed to assess the repeated addition of CSS on soil fertility in the long term.

**Author Contributions:** Conceptualization, V.A.L. and E.P.; methodology, V.A.L. and E.P.; formal analysis, C.L. and D.P.; investigation, C.L. and D.P.; data curation, C.L. and D.P.; writing—original draft preparation, V.A.L., L.B., E.P., C.L. and D.P.; writing—review and editing, V.A.L., L.B., E.P., C.L. and D.P.; supervision, V.A.L. All authors have read and agreed to the published version of the manuscript.

**Funding:** This research received no external funding.

**Institutional Review Board Statement:** Not applicable.

**Data Availability Statement:** The data presented in this study are available upon request from the corresponding author.

**Acknowledgments:** The authors thank Anna Micalizzi for her support during soil analyses.

**Conflicts of Interest:** The authors declare no conflict of interest.

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
