# Peer review of "From Waste to Resources: Sewage Sludges from the Citrus Processing Industry to Improve Soil Fertility and Performance of Lettuce (Lactuca sativa L.)"

_agriculture, doi:10.3390/agriculture13040913_

Round 1
Reviewer 1 Report
The aim of this study was to assess the effect of CSSs supplied at different doses on soil fertility and lettuce performance. To this end, a pot experiment was established. Soil was amended with CSSs at three different con-centrations (2.5, 5, 10 t ha-1). After 46 days of lettuce growth the experiment was stopped, and soils and plants were analysed. Soil amended with CSSs showed an increase of total organic C ranging from 7% to 11%. Also available P increased but only at the highest CSSs dose. The addition of CSSs affected also the biochemical properties of soil, but univocal trend related to the amount of CSSs applied was not found. Microbial biomass C increased only with the highest dose of CSSs applied, while the metabolic quotient (qCO2) decreased. Such a positive effect on soil fertility and soil microorganisms, in turn, lead to an increase of lettuce biomass.
Overall, the manuscript looks nice, some minor changes are suggested for the improvement.
1. Conclusions should be reduced below 200 words.
2. I anticipated the work on soil microbial biomass consequent to sludge application, can authors add any data if they have?
3. Another feature which author needed to study was chlorophyll contents separately, any data on this?
4. Please write QA/QC exercised during the experiment.
Author Response
REVIEWER#1
Dear Reviewer,
on behalf of all the Co-Authors, I would like to thank you for the suggestions and comments to our Ms. We accepted all the suggested changes and answered to yours comments. We really appreciate your suggestions that push us to improve the manuscript. Please find below the detailed reply to each comment (in black we reported your comments, in red our answer).
Sincerely
The Corresponding Author
Daniela Pampinella
The aim of this study was to assess the effect of CSSs supplied at different doses on soil fertility and lettuce performance. To this end, a pot experiment was established. Soil was amended with CSSs at three different con-centrations (2.5, 5, 10 t ha-1). After 46 days of lettuce growth the experiment was stopped, and soils and plants were analysed. Soil amended with CSSs showed an increase of total organic C ranging from 7% to 11%. Also available P increased but only at the highest CSSs dose. The addition of CSSs affected also the biochemical properties of soil, but univocal trend related to the amount of CSSs applied was not found. Microbial biomass C increased only with the highest dose of CSSs applied, while the metabolic quotient (qCO2) decreased. Such a positive effect on soil fertility and soil microorganisms, in turn, lead to an increase of lettuce biomass.
Overall, the manuscript looks nice, some minor changes are suggested for the improvement.
Authors answer: The authors thank the Reviewer for his/her appreciation of the study.
- Conclusions should be reduced below 200 words.
Authors answer: Conclusions paragraph is of 176 words.
- I anticipated the work on soil microbial biomass consequent to sludge application, can authors add any data if they have?
Authors answer: The Authors thank the Reviewer for the suggestion. Effect of sewage sludge on soil microbial biomass were added in the Introduction.
- Another feature which author needed to study was chlorophyll contents separately, any data on this?
Authors answer: data about chlorophyll are reported in figure 3 (SPAD index).
- Please write QA/QC exercised during the experiment.
Authors answer. The Authors thank the Reviewer for the suggestion. QA/QC measures and procedures were added at the end of each paragraph of the materials and methods section.
Reviewer 2 Report
This study describes a study that aimed to evaluate the potential use of citrus sewage sludges (CSSs) as organic fertilizers for agricultural purposes, due to their high levels of nitrogen, phosphorus, and other macronutrients required by crops. The study used a pot experiment to assess the effect of different concentrations of CSSs on soil fertility and lettuce performance. The results showed that amending soil with CSSs led to an increase in total organic C and available P, as well as an improvement in soil microbial biomass and lettuce biomass production, without affecting its quality, which is a circular economy approach to promoting waste management. The manuscripts are rich in content, meet the requirements of journals, and have good analytical means. I suggest that the manuscript be accepted after major revision.
1. The basic parameters of the sludge feedstock require an additional indicator for heavy metals to confirm that the use is not risky.
2. Are there any limitations or challenges associated with utilizing CSSs for agricultural reuse that were identified by the authors? I suggest additional discussion.
3. The data inside the table needs to have error values, please add them.
4. It is recommended to add a description of other resourceful applications of sludge, such as preparation into biochar to improve soil nutrient properties, remediation of heavy metals in soil and protection of food security. Please refer to: https://doi.org/10.3390/agronomy12051003; https://doi.org/10.1016/j.envint.2019.03.068; https://doi.org/10.3389/fbioe.2023.1127166
Author Response
REVIEWER#2
Dear Reviewer,
on behalf of all the Co-Authors, I would like to thank you for the suggestions and comments to our Ms. We really appreciate your suggestions that push us to improve the manuscript. Please find below the detailed reply to each comment (in black we reported your comments, in red our answer).
Sincerely
The Corresponding Author
Daniela Pampinella
This study describes a study that aimed to evaluate the potential use of citrus sewage sludges (CSSs) as organic fertilizers for agricultural purposes, due to their high levels of nitrogen, phosphorus, and other macronutrients required by crops. The study used a pot experiment to assess the effect of different concentrations of CSSs on soil fertility and lettuce performance. The results showed that amending soil with CSSs led to an increase in total organic C and available P, as well as an improvement in soil microbial biomass and lettuce biomass production, without affecting its quality, which is a circular economy approach to promoting waste management. The manuscripts are rich in content, meet the requirements of journals, and have good analytical means. I suggest that the manuscript be accepted after major revision.
- The basic parameters of the sludge feedstock require an additional indicator for heavy metals to confirm that the use is not risky.
Authors answer: The authors appreciate the suggestion of the Reviewer although no indication is provided about the additional indicator of which heavy metals are missing. The parameters reported in table 1 are those required by the Directive 86/278/ECC. The only missing heavy metal according to such a Directive is Hg. The determination of Hg is not possible by facilities available in our laboratory. However, the citrus industry periodically checks for Hg content. Following such controls, never Hg has been detected.
- Are there any limitations or challenges associated with utilizing CSSs for agricultural reuse that were identified by the authors? I suggest additional discussion.
Authors answer: The authors thank the reviewer for the suggestion. Based on the results reported in this paper and the lack of data in the literature on CSS, we do not believe, at this stage, that there are any particular limitations or challenges in their reuse in agriculture. However, it would be desirable further investigation, especially field trials, to understand the quantity of CSS beyond which their application could become an issue.
- The data inside the table needs to have error values, please add them.
Authors answer: The authors thank the reviewer for the suggestion. However, we think that the use of letters to indicate significant differences among treatments is exhaustive. The addition of standard deviations did not add any further information. Moreover, by checking other published papers on the same Journal we found that it is not mandatory to add standard deviations.
- It is recommended to add a description of other resourceful applications of sludge, such as preparation into biochar to improve soil nutrient properties, remediation of heavy metals in soil and protection of food security. Please refer to: https://doi.org/10.3390/agronomy12051003; https://doi.org/10.1016/j.envint.2019.03.068; https://doi.org/10.3389/fbioe.2023.1127166
Authors answer: The Authors thank the Reviewer for the suggestion. Others resourceful application of the sludge were added in the introduction also on the basis of the suggested bibliography.
Reviewer 3 Report
Overall, some parts of this article need to be adjusted.
(1) The introduction section can be appropriately reduced in length and paragraphs can be redistributed.
(2) The tools and methods section can be appropriately expanded and detailed, such as adding illustrations.
(3) The results in Table 1 and Figure 2 and Figure 3 are too simple and need to be redone.
Author Response
REVIEWER#3
Dear Reviewer,
on behalf of all the Co-Authors, I would like to thank you for the suggestions and comments to our Ms. We accepted all the suggested changes and answered to yours comments. We really appreciate your suggestions that push us to improve the manuscript. Please find below the detailed reply to each comment (in black we reported your comments, in red our answer).
Sincerely
The Corresponding Author
Daniela Pampinella
Overall, some parts of this article need to be adjusted.
Authors answer: The Authors thank the Reviewer for the suggestions/comments provided.
(1) The introduction section can be appropriately reduced in length and paragraphs can be redistributed.
Authors answer: Introduction was revised also according to the suggestions provided by Reviewer 1 and 2.
(2) The tools and methods section can be appropriately expanded and detailed, such as adding illustrations.
Authors answer: The Authors thank the Reviewer for the suggestions. More details were added to the M&M section.
(3) The results in Table 1 and Figure 2 and Figure 3 are too simple and need to be redone.
Authors answer: Standard deviation in table 1 was added and captions of figure 2 and 3 improved.
Round 2
Reviewer 2 Report
1. The error value of the data is an indication of the authenticity of the data and from my personal point of view should be increased. 2. As and Hg in the raw material if not available please specify in the method.
Author Response
Answer to Reviewer 2 comments
Reviewer Comment: The error value of the data is an indication of the authenticity of the data and from my personal point of view should be increased.
Answer: the Authors thank the Reviewer for the suggestion. Standard deviations were added.
Reviewer Comment: As and Hg in the raw material if not available please specify in the method.
Answer: done.
Reviewer 3 Report
The chart can still be optimized.
Author Response
Answer to Reviewer 3 comments
Reviewer Comment: The chart can still be optimized.
Answer: done. Figures were improved by using the same font of the manuscript.